# Stable salts of the hexacarbonyl chromium(I) cation and its pentacarbonyl-nitrosyl chromium(I) analogue

Jan Bohnenberger [1], Wolfram Feuerstein[2], Daniel Himmel[1], Michael Daub[1], Frank Breher [2] & Ingo Krossing[1]

Homoleptic carbonyl radical cations are a textbook family of complexes hitherto unknown in the condensed phase, leaving their properties and applications fundamentally unexplored. Here we report on two stable 17-electron $[Cr(CO)_6]^{\bullet+}$ salts that were synthesized by oxidation of $Cr(CO)_6$ with $[NO]^+[Al(OR^F)_4]^-$ ($R^F = C(CF_3)_3$)) in $CH_2Cl_2$ and with removal of NO gas. Longer reaction times led to NO/CO ligand exchange and formation of the thermodynamically more stable 18-electron species $[Cr(CO)_5(NO)]^+$, which belongs to the family of heteroleptic chromium carbonyl/nitrosyl cations. All salts were fully characterized (IR, Raman, EPR, NMR, scXRD, pXRD, magnetics) and are stable at room temperature under inert conditions over months. The facile synthesis of these species enables the thorough investigation of their properties and applications to a broad scientific community.

[1] Institut für Anorganische und Analytische Chemie and Freiburger Materialforschungszentrum (FMF), Universität Freiburg, Albertstr. 21, 79104 Freiburg, Germany. [2] Karlsruhe Institute of Technology (KIT), Division Molecular Chemistry, Institute of Inorganic Chemistry, Engesserstr. 15, 76131 Karlsruhe, Germany. Correspondence and requests for materials should be addressed to F.B. (email: breher@kit.edu) or to I.K. (email: krossing@uni-freiburg.de)

Carbon monoxide is amongst the most important ligands in transition metal chemistry. Thus, carbonyl complexes have been continuously and extensively studied over the past 130 years after the discovery of the first homoleptic carbonyl complex $Ni(CO)_4$ by Mond in 1890[1]. Their beauty lies in their simplicity and their rich substitution and redox chemistry, which leads to a wide range of applications, spanning from catalysis to biochemistry and medicine[2–7]. Most condensed phase homoleptic transition metal carbonyl complexes and all neutral mononuclear homoleptic transition metal carbonyl complexes in particular, obey the 18-electron rule[7]; the only exception is $V(CO)_6$ as a 17 valence electron (VE) species. Neutral complexes can be reduced to gain anionic carbonyl metallates, such as $[V(CO)_6]^-$, $[Fe(CO)_4]^{2-}$ (discovered as early as 1930) or even clustered radical anions such as $[Fe_3(CO)_{11}]^{\bullet-}$[8,9]. Transition metal carbonyl cations (TMCCs), however, could not be accessed until about 1960, when the octahedral carbonyl cation $[Mn(CO)_6]^+$ was discovered[10]. Since the 1990s, mainly superacidic media enabled the synthesis, isolation, and characterization of several homoleptic TMCCs, such as $[Au(CO)_2]^+$[11–13], $[Fe(CO)_6]^{2+}$[14], $[Co(CO)_5]^+$[15] or even superelectrophilic $[Pd(CO)_4]^{2+}$[16], or $[Ir(CO)_6]^{3+}$[17]. Despite the fact that all these complexes were prepared by an oxidative approach, all of them feature an even number of electrons. No open-shell TMCC with mononuclear metalloradical configuration[18] has been synthesized and structurally characterized to date (cf. Table 1)[19–29]. Typically, those would evade the open-shell situation by forming metal–metal-bonded dimers, like seen for the reliably on spectroscopic data assigned dimeric platinum carbonyl cation $[(OC)_3Pt–Pt(CO)_3]^{2+}$[23].

However, the chromium hexacarbonyl cation, as a prototype example for such open-shell TMCC systems, was the subject of several electrochemical investigations[30–33], as well as gas phase and theoretical studies[34–36]. In principle, these rather weakly bound TMCCs should be ideal starting materials for further substitution chemistry. Yet, the established routes to TMCCs via superacidic media require specialized equipment/laboratories and superacids are often incompatible with commonly used ligands. This combination limited hitherto the possibilities for follow-up chemistry. Quite contrary, the approach delineated here uses the weakly coordinating anions (WCAs) $[Al(OR^F)_4]^-$ and $[F-\{Al(OR^F)_3\}_2]^-$ $(R^F = C(CF_3)_3)$ that allow the use of regular organic solvents, as well as standard laboratory equipment. In combination, this approach facilitates synthesis of the target TMCC salts—if desired, also in multigram-scale—and enables a rich follow-up chemistry.

## Results and Discussion

**Synthesis**. A suitable oxidant (strong and innocent) is required to prepare the target salts $[Cr(CO)_6]^{\bullet+}[WCA]^-$. Whilst $Ag^+$ $(E° = +0.65\,V$ vs. $Fc/Fc^+$ in $CH_2Cl_2)$[37] deelectronates $[Co_2(CO)_8]$ in

CO atmosphere generating $[Co(CO)_5]^+$ at room temperature (r.t.) in ortho-difluorobenzene $(oDFB)$[38], its redox potential is too low to oxidize $Fe(CO)_5$. This leads to the formation of the peculiar metal-only Lewis pair $[Ag\{Fe(CO)_5\}_2]^+$[39,40]. $[NO]^+$ might be a strong enough oxidant (cf. $E° = +1.00\,V$ vs. $Fc/Fc^+$ in $CH_2Cl_2)$[37], but it is prone to an inner-sphere electron transfer, which then often leads to a coordination of the resulting NO to the metal center[37,41]. This is also reflected in the reaction of $[Co_2(CO)_8]$ with $[NO]^+$ that yielded an inseparable mixture of $[Co(CO)_5]^+$ and $[Co(CO)_2(NO)_2]^+$[42]. However, the octahedral configuration of $Cr(CO)_6$ should be kinetically hindered towards substitution due to an energetically disfavored mechanism[43]. This feature enabled using $[NO]^+$ as oxidant in combination with the WCAs $[Al(OR^F)_4]^-$ and $[F-\{Al(OR^F)_3\}_2]^-$. In order to minimize the inevitable NO/CO-ligand exchange, the 1:1 deelectronation reaction of $Cr(CO)_6$ with $[NO]^+[WCA]^-$ (WCA: $[Al(OR^F)_4]^-$ or $[F-\{Al(OR^F)_3\}_2]^-$) was carried out at low temperatures in $CH_2Cl_2$ (−78 °C to r.t.). Application of a dynamic vacuum over 1 h in order to immediately remove the released NO, furnished near quantitative yields of the 17 VE species $[Cr(CO)_6]^{\bullet+}[Al(OR^F)_4]^-$ (1) or $[Cr(CO)_6]^{\bullet+}[F-\{Al(OR^F)_3\}_2]^-$ (2) as off-white to pale-yellow solids (Eq. 1). By allowing the reaction to proceed for 7–14 days in a closed vessel at r.t. (Eq. 2), the heteroleptic 18 VE complexes $[Cr(CO)_5(NO)]^+[Al(OR^F)_4]^-$ (3) and $[Cr(CO)_5(NO)]^+[F-\{Al(OR^F)_3\}_2]^-$ (4) were obtained in near quantitative yield as orange solids. Here, the liberated NO displaced one CO ligand and acts as a $3e^-$ donor (Fig. 1).

The success of the reaction was monitored by gas-phase IR spectroscopy. The loss of either CO or NO was observed in the reactions providing 3 and 4 or 1 and 2, respectively (see S.I. section 8, Supplementary Figures 35 and 36). The purity and phase homogeneity of 1–4 were confirmed by IR and Raman spectroscopy, as well as by phase analysis with powder X-ray diffraction measurements (pXRD, see S.I. section 13, Supplementary Figures 51–58). All complexes are stable at room temperature and can be stored for months under argon atmosphere. In addition, magnetic measurements using the Evans method[44] in $oDFB/CH_2Cl_2$ solutions furnished the expected magnetic moments of $2.04\mu_B$ for 1 and $2.06\mu_B$ for 2, respectively (see S.I. section 6, Supplementary Table 1, Supplementary Figures 27–32). Hence, apart from being a powerful method for the synthesis of TMCCs, Eq. 2 also opened the door to the family of heteroleptic chromium carbonyl/nitrosyl cations. To put this into perspective, $[Cr(CO)_5(NO)]^+$ has only been subject to a few theoretical studies yet[45,46] and structurally authenticated examples are only known for anionic $([Cr_2(CO)_9(NO)]^-$, $[Cr(CO)_4(NO)]^-)$[47–49] or neutral $(Cr_2(CO)_8(NO)_2)$[47] representatives.

**Kinetic vs. thermodynamic product**. From a series of reactions performed at intermediate temperatures and monitored at

**Table 1 Currently known and structurally characterized homoleptic transition metal carbonyl cations of group 7–12[a]**

| | 7 | 8 | 9 | 10 | 11 | 12 |
|---|---|---|---|---|---|---|
| 3d | $[Mn(CO)_6]^+$[24] | $[Fe(CO)_6]^{2+}$ | $[Co(CO)_5]^+$ | – | $[Cu(CO)_n]^+$ $(n = 1-4)$[25-27] | – |
| 4d | $[Tc(CO)_6]^+$[28] | $[Ru(CO)_6]^{2+}$ | $[Rh(CO)_4]^{+\,b}$ | $[Pd(CO)_4]^{2+\,b}$ $[Pd_2(\mu-CO)_2]^{2+\,c}$ | $[Ag(CO)_n]^+$ $(n = 1-2)$[25,27,29] | – |
| 5d | $[Re(CO)_6]^+$ | $[Os(CO)_6]^{2+}$ | $[Ir(CO)_4]^{+\,b}$ $[Ir(CO)_6]^{3+}$ | $[Pt(CO)_4]^{2+\,b}$ | $[Au(CO)_2]^+$ | $[Hg(CO)_2]^{2+}$ $[Hg_2(CO)_2]^{2+}$ |

[a]From Group 3–6, no such cations are known. Only structurally characterized examples are shown; transition metal carbonyl cations are only referenced, if not mentioned in the reviews[19–22]
[b]Sixteen valence electron species
[c]Bidentate, weakly bridging fluorosulfate groups are present

various times, it appeared that the oxidation and formation of **1** and **2** is the kinetically preferred reaction pathway, whereas the nitrosyl complexes **3** and **4** are the thermodynamic final products. Full ab initio calculations at the CCSD(T)/TZ→QZ level (see S.I. section 16, Supplementary Figure 75) support the claim that $[Cr(CO)_6]^{\bullet+}$ is indeed the kinetic product in $CH_2Cl_2$ as solvent $(\Delta_r H^{\circ}_{gas}/\Delta_r G^{\circ}_{solv} = -83/-30 \, kJ \, mol^{-1}$ for $[Cr(CO)_6]^{\bullet+}$ (Eq. 1) vs. $-232/-173 \, kJ \, mol^{-1}$ for $[Cr(CO)_5NO]^+$, Eq. 2). The high reactivity of $[NO]^+$ and rather poor solubility of **1** and **2** in $CH_2Cl_2$ seem to be crucial for the success of the reaction (Eq. 1), since it leads to a quick formation of the relatively stable

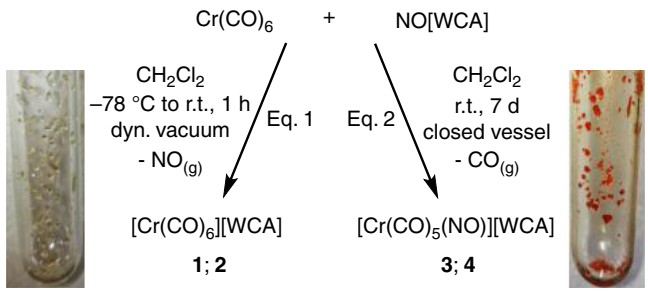

$[Cr(CO)_6]^{\bullet+}$ intermediate that shortly after precipitates at low temperatures. By contrast, carrying out the reaction in *o*DFB, where all the products are well dissolved, apparently led to a less reactive $[NO]^+$ and therefore only the isolation of $[Cr(CO)_5(NO)]^+$. We note that the reaction of the obviously formed charge-transfer π−complex $[NO^+$-*o*DFB$]$ with $Cr(CO)_6$ must be faster than the formation of a Wheland-intermediate and nitrosation of the solvent. However, once pure **3** and **4** were prepared, their crystals dissolve unchanged in *o*DFB.

**Vibrational spectroscopy**. The solid state ATR IR spectra (Fig. 2 and Table 2; ATR = attenuated total reflection) of **1** and **2** show a single broadened CO band at $2094 \, cm^{-1}$ (small $^{13}CO$ shoulder at about $2070 \, cm^{-1}$), which verifies the assumptions made of a possible $[Cr(CO)_6]^{\bullet+}$ in a previous matrix study[50]. The respective Raman spectra show a sharp band at about $2174 \, cm^{-1}$ for the all-symmetric stretch vibration, as well as a broad band centered around $2127 \, cm^{-1}$. The position and shape of these bands are in agreement with the Raman spectrum of the isoelectronic but neutral $D_{3d}$-symmetric $V(CO)_6$ (cf. Table 2 and Supplementary Figure 34)[51].

A Jahn–Teller-induced fluxionality at room temperature leads to the broad band at about $2127 \, cm^{-1}$ not affecting the all-symmetric stretching mode. This fluxionality freezes out in the vanadium case only at temperatures below 16 K. At higher temperatures, a very low-lying transition state probably allows for equilibration—even

**Fig. 1** Reaction scheme yielding complexes **1** and **2** (Eq. 1) as well as **3** and **4** (Eq. 2). The pictures show the crystalline complexes **1** (pale yellow) and **3** (orange)

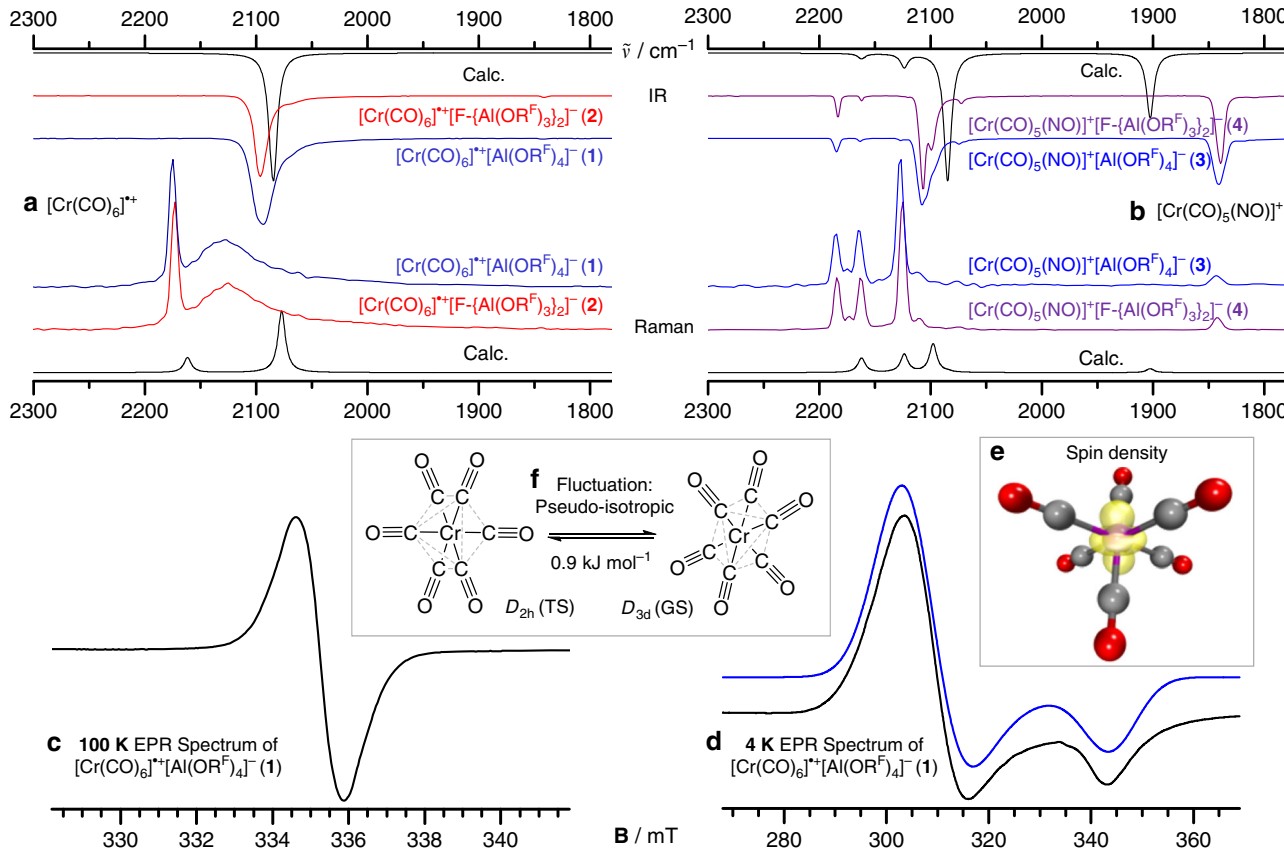

**Fig. 2** Block figure showing essential IR, Raman, and EPR spectra of complexes **1**–**4**. **a**, **b** Stacked IR and Raman spectra of compounds **1** (dark blue), **2** (red), **3** (blue), **4** (purple), and the respective simulated calculated spectra (black, BP86-D3BJ/def2-TZVPP) of the cations in the CO/NO stretching range between 1800 and 2300 $cm^{-1}$; For the full spectra, including an assignment of all bands see Supplementary Figures 42–49 and Supplementary Tables 2, 3. **c** EPR spectrum of **1** at X-Band (9.4002 GHz) in a frozen solution of a 1:1 (v:v) mixture of *o*DFB and toluene at 100 K. **d** Experimental (black) and simulated (blue) EPR spectrum of **1** at X-Band (9.37821 GHz) in a frozen solution of *o*DFB at 4 K in agreement with the $D_{3d}$ ground state. **e** Calculated spin density of $D_{3d}$-$[Cr(CO)_6]^{\bullet+}$ at the SA-CASSCF/cc-pVTZ level of theory. Isovalue at 0.02 a.u. **f** Equilibration path that transforms the two equivalent $D_{3d}$ ground states (GS) at 100 K over a low-lying $D_{2h}$ transition state (TS), yielding a coalescent signal in the 100 K EPR spectrum

**Table 2 Experimental (exp.) and calculated (calcd.) vibrational spectra of $[Cr(CO)_6]^{\bullet+}$, $[Cr(CO)_5(NO)]^+$, and the isoelectronic vanadium analogs$^a$**

| IR Spectra | | | Raman spectra | | | | IR$^b$ | Raman$^c$ |
|---|---|---|---|---|---|---|---|---|
| Calcd. ($I$)$^d$ | 1 | 2 | 1 | 2 | Calcd. ($I$)$^d$ | $D_{3d}$ | Exp. V(CO)$_6$[51] | |
| 2157 (0) | | | 2175 (s) | 2173 (s) | 2157 (208) | $A_{1g}$ | | 2102 (12.5) |
| 2084 (816) | 2094 (s) | 2096 (s) | | | 2084 (0) | $A_{2u}$ | 1991 (54) | |
| 2081 (999) | | | | | 2081 (0) | $E_u$ | 1985 (100) | |
| 2074 (0) | | | 2128 (br, vs) | 2126 (br, vs) | 2074 (413) | $E_g$ | | ≈1970 (vbr, vs)$^c$ |
| Calcd. ($I$)$^d$ | 3 | 4 | 3 | 4 | Calcd. ($I$)$^d$ | $C_{4v}$ | Exp. V(CO)$_5$(NO)[52] | Not available |
| 2161 (91) | 2184 (vw) | 2183 (vw) | 2185 (m) [2175 (vw)]$^e$ | 2184 (m) [2173 (vw)]$^e$ | 2161 (177) | $A_1$ | 2100 (w) | |
| 2123 (91) | 2164 (vvw) | 2162 (vvw) | 2164 (m) | 2163 (m) | 2123 (217) | $A_1$ | 2050 (w) | |
| 2097 (0) | 2127 (vvw) | 2127 (vvw) | 2127 (vvs) | 2125 (vvs) | 2097 (340) | $B_2$ | | |
| 2085 (1055) | 2108 (ms) | 2107 (vs) 2099 (m)$^f$ | 2112 (vw) | 2110 (vvw) | 2085 (5) | $E$ | 1990 (s) | |
| | | | | | | $E$ | | |
| | 2074 (vvw)$^g$ | 2072 (vvw)$^g$ | 2076 (vvw)$^g$ | 2074 (vvw)$^g$ | | $^{13}$CO$^g$ | | |
| 1899 (1052) | 1841 (mw) | 1839 (ms) | 1843 (vvw) | 1842 (vvw) | 1899 (39) | $A_1$ NO | 1695 (m) | |

$^a$IR and Raman bands and relative intensities of the CO and NO vibrations are given for complexes **1–4** and the DFT calculated (BP86-D3BJ/def2-TZVPP) values for the $[Cr(CO)_6]^{\bullet+}$ and $[Cr(CO)_5(NO)]^+$ cations; (br: broad, v: very, s: strong, m: medium, w: weak); Raman and IR intensities were calculated with the Gaussian software. Note that here we gave preference to the BP86 functional, which gave better unscaled absolute band positions than the TPSSh functional chosen for bond lengths and EPR calculations (vide infra and Supplementary Table 4)
$^b$At 5 K in Neon matrix[51].
$^c$At room temperature in benzene solution. The signal is very broad and covers a wide range from 2050 to 1850 cm$^{-1}$
$^d$IR-intensities in km mol$^{-1}$ and Raman scattering activities in Å$^4$ amu$^{-1}$
$^e$Small contamination with $A_{1g}$-stretch of $[Cr(CO)6]^{\bullet+}$ in **1** (2175 cm$^{-1}$) and **2** (2173 cm$^{-1}$)
$^f$Probably removal of the degeneracy of the $E$-mode in **4** (2107 cm$^{-1}$). In **3** the signal is broader and covers this detail
$^g$Probably the isotope shifted $^{13}$CO band of the intense $E$-mode at 2108/2107 (IR) and 2112/2110 (Raman) cm$^{-1}$. Exp. $\Delta(\nu(^{12}CO) - \nu(^{13}CO)) = 34$ (**3**), 33 (**4**) cm$^{-1}$ for IR; 36 (**3**), 36 (**4**) for Raman. Calcd.: 36 cm$^{-1}$ for an explicit $^{13}$CO ligand placed in equatorial position ($C_s$-symmetry; absolute calculated position: 2048 cm$^{-1}$ ($I$: 735))

**Table 3 Comparison of experimental (exp.) and theoretical (calcd.) bond lengths of 1–4 and selected literature complexes**

| | Exp. bond | Lengths (pm)$^a$ | | | Exp. bond | Lengths (pm)$^b$ | Calcd. bond | Lengths (pm)$^c$ | |
|---|---|---|---|---|---|---|---|---|---|
| | $[Al(OR^F)_4]^-$ | | $[F\text{-}\{Al(OR^F)_3\}_2]^-$ | | | | $[Cr(CO)_6]^+$ | $[Cr(CO)_5(NO)]^+$ | Cr(CO)$_6$ |
| | $[Cr(CO)_6]^+$ | $[Cr(CO)_5(NO)]^+$ | $[Cr(CO)_6]^+$ | $[Cr(CO)_5(NO)]^+$ | Cr(CO)$_6$[53] | V(CO)$_6$[54] | $D_{3d}$ | $C_{4v}$ | $O_h$ |
| Cr-C1 | 196.2(3) | 195.8(3) | 198.2(2) | 194.9(4) | 191.4(1) | 199.3(5) | 197.5 | 202.0 | 190.9 |
| Cr-C2$^d$ | 199.1(1) | 196.2(1) | | | 191.1(1)$^e$ | 199.4(5)$^e$ | | 195.8 | |
| Cr-C3 | 196.9(3) | 192.4(3) | | | 190.9(1) | 200.3(4) | | 172.9 (Cr-N) | |
| | | | | | 191.7(1)$^e$ | 200.6(4)$^e$ | | | |
| C1-O1 | 112.4(3) | 112.4(3) | 112.2(2) | 112.5(5) | 114.2(1) | 113.3(4) | 113.0 | 112.6 | 114.4 |
| C2-O2$^d$ | 112.0(1) | 112.5(2) | | | 114.2 (1)$^e$ | 113.9(4)$^e$ | | 113.1 | |
| C3-O3 | 112.4(3) | 113.0(3) | | | 114.4(1) | 113.3(3) | | 113.6 (N-O) | |
| | | | | | 113.9(1)$^e$ | 112.4(3)$^e$ | | | |
| Avg.$^f$ Cr-C | 198.6(4) | 195.6(4) | 198.2(2) | 194.9(4) | 191.3(2) | 199.9(9) | 197.5 | 193.0 | 190.9 |
| Avg.$^f$ C-O | 112.1(4) | 112.6(5) | 112.2(2) | 112.5(5) | 114.1(2) | 113.2(7) | 113.0 | 113.1 | 114.4 |

$^a$This work
$^b$From literature
$^c$TPSSh-D3BJ/def2-TZVPP
$^d$Ligand on the $C_4$-symmetry axis
$^e$$C_2$-symmetry axis
$^f$Mean error of bond lengths were calculated according to Gaussian error propagation

on the fast time scale of vibrational spectroscopy, leading to the discrepancy between experimental and the $D_{3d}$-simulated spectrum. This assignment is in agreement with the EPR spectra as well as DFT and ab initio calculations for the cation (vide infra, Table 3). The presence of a band of very low intensity at about 1840 cm$^{-1}$ is indicative for the formation of very small amounts of the (by-)products $[Cr(CO)_5(NO)]^+$ **3** and **4**. Furthermore, as opposed to **1** and **2**, some Cr–N and Cr–C vibrations at 658/640 cm$^{-1}$ (IR) and 640/487 cm$^{-1}$ (Raman) are not obscured by anion bands and can be assigned unambiguously (cf. Supplementary Figure 33). Their entire vibrational spectra are in good agreement with the simulation (Fig. 2), as well as the known bands of the isoelectronic neutral V(CO)$_5$(NO)[52] (Table 2).

**Molecular structures.** Quite unsurprisingly and due to their very similar molecular structure, the pale yellow complexes **1**, **2** and the orange colored complexes **3**, **4** are isostructural. The high symmetry of the crystal structures of **1** and **3** (tetragonal, $P4/n$) leads to only three symmetry-independent ligands, rendering a differentiation between the NO or CO positions impossible. Even more so for **2** and **4**, where the cubic structures ($Pa\bar{3}$) include only one symmetry-independent ligand (Supplementary Figure 68 and Supplementary Figure 71). Yet, spectroscopically clean pale yellow colored (orange colored) crystals suitable for single crystal XRD (scXRD), in agreement with an assignment as **1** and **2** (**3** and **4**), were isolated from batches of reactions according to Eq. 1 (Eq. 2) (Fig. 3).

The observed average Cr–ligand bond lengths are about 3 pm shorter for **3** and **4**. This is consistent with the shorter (calculated) Cr–N bond length (Table 3) and is in addition reflected in smaller unit cell volumes of **3** and **4**, if compared to **1** and **2**. In addition, this also holds for the bulk microcrystalline powders of all compounds **1–4**, as shown by full Rietveld-refinement of powder XRD data collected at 100 K (see S.I. section 13, Supplementary Table 7). Overall, the determined experimental parameters and the

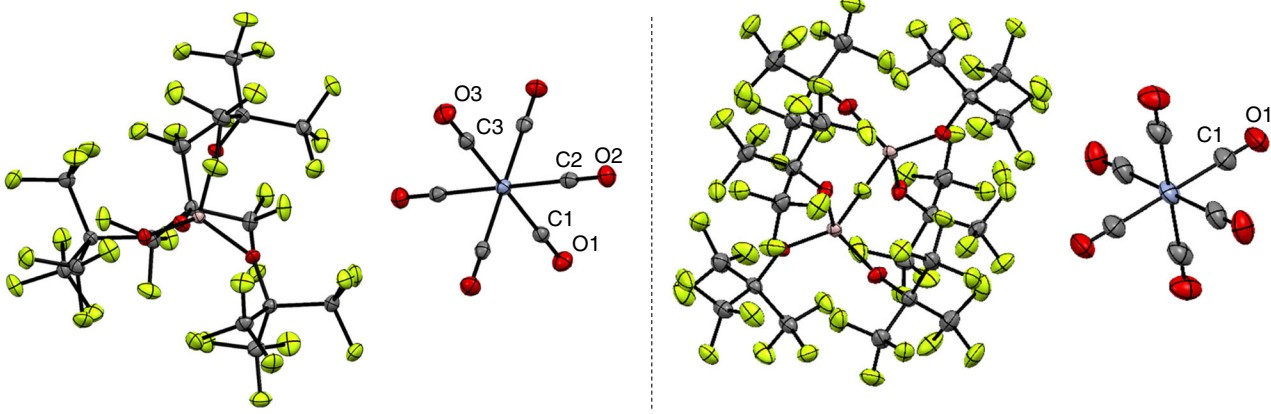

**Fig. 3** Molecular structures of complexes **1–4**. Pale yellow colored (orange colored) crystals suitable for scXRD and with spectroscopic data in agreement with an assignment as **1** and **2** (**3** and **4**) were isolated from batches of reactions according to Eq. 1 (Eq. 2). However, the respective nitrogen position was never crystallographically distinguishable from the carbon positions. Therefore, only one set of molecular structures is shown: **1** and **3** (left) and **2** and **4** (right). Inclusion of a disordered nitrogen atom into the refinement did only slightly change the agreement factors (Supplementary Figures 70 and 72). Nevertheless, the average bond lengths are about 3 pm shorter for **3** and **4**, which agrees with the (calculated) shorter Cr–N bond length and the observed smaller unit cell volumes. Only crystallographically independent atoms were labeled; ellipsoids are shown at 50% probability level. See Table 3 for information on metric data

results of DFT calculations collected in Table 3 are in good agreement and in accordance with the reported bond lengths for Cr$(CO)_6$[53] and V$(CO)_6$[54].

**EPR spectroscopy.** To verify the identity of the $[Cr(CO)_6]^{\bullet+}$ radical cation, a diluted frozen solution of **1** in a 1:1 (v:v) mixture of oDFB and toluene (pure oDFB solutions led to very noisy spectra) was investigated by means of X-band EPR spectroscopy at 100 K (Fig. 2c). The paramagnetic 17 VE $d^5$ species gives rise to one single EPR-line corresponding to an isotropic $g$-value of $g = 2.003$ being close to the $g$-factor of the free electron ($g_{el} = 2.0023$)[55]. Similar $g$-values were reported for tetra-, tri-, and dicarbonyl chromium(I) phosphine and phosphite complexes[56]. In contrast to prior EPR investigations from electrochemical oxidations[33], we did not observe any signal splitting due to hyperfine coupling to the $^{53}$Cr nucleus ($I = 3/2$, natural abundance: 9.55%). Probably any coupling pattern is hidden by the line broadening in the EPR spectra, since our calculations of hyperfine couplings (HFCs) reveal isotropic coupling constants always lower than 30 MHz (Supplementary Table 5). Note, however, that a larger hyperfine coupling of 61 MHz was reported for a species assigned as $[Cr(CO)_6]^{\bullet+}$ and generated by in situ electrolysis of Cr$(CO)_6$ in saturated solutions of $[NBu_4]^+[ClO_4]^-$ in $CH_2Cl_2$ at $-80\,°C$[33]. This $^{53}$Cr hyperfine coupling is about three times larger than the coupling constants of chromium(I) carbonyl phosphine and phosphite complexes[56]. Furthermore, the $g$-value of $g = 2.003$ we identified for **1** is larger than the one previously reported ($g = 1.982$)[33]. We note, however, that the presence of the—compared to our WCAs—much stronger coordinating perchlorate anion in the experimental setup may have led to ion-pair formation, thus shifting the $g$-value with respect to the naked $[Cr(CO)_6]^{\bullet+}$ cation. Our isotropic $g$-value obtained at 100 K would be consistent with an octahedral symmetry $O_h$ of the $[Cr(CO)_6]^{\bullet+}$ cation. However, the $d^5$ complex should be subject to a dynamical Jahn–Teller effect, lowering its symmetry as it was found in the crystal structure of **1** and **2** and that of the isoelectronic V$(CO)_6$, which has $D_{3d}$ symmetry in its electronic ground state[51] (see S.I. section 9 for an in-depth discussion). Consequently, the pseudo-isotropic signal at 100 K may result from a thermal fluctuation of different Jahn–Teller distorted $[Cr(CO)_6]^{\bullet+}$ structures (e.g., the $D_{3d}$–$D_{2h}$–$D_{3d}$ sequence shown in Fig. 2f and also suggested for V$(CO)_6$[51]). To verify this

**Table 4 Calculated relative electronic energies $E$ and $g$-values with different point groups for the $[Cr(CO)_6]^{\bullet+}$ cation[a]**

|  | Exp. | $D_{3d}$ | $C_{2h}$ | $D_{4h}$ | $D_{2h}$[b] | $O_h$[c] |
|---|---|---|---|---|---|---|
| $\Delta E$ |  | 0 (0.00) | 26 (0.31) | 136 (1.63) | 71 (0.85) | 410 (4.92) |
| $g_\perp$ | 2.185 | 2.173 | 2.177 | 2.434 |  |  |
| $g_\parallel$ | 1.947 | 1.971 | 1.969 | 1.761 |  |  |
| $g_{iso}$[d] | 2.106 | 2.106 | 2.108 | 2.210 |  |  |

[a]Electronic energies were calculated with DLPNO-CCSD(T)/def2-TZVPP ($E$ in cm$^{-1}$; kJ mol$^{-1}$ in parentheses), structures were optimized with TPSSh-D3BJ/def2-TZVPP; the calculated (NEVPT2-SA-CAS-SCF/cc-pVTZ) anisotropic $g$-tensor components perpendicular ($g_\perp$) and parallel ($g_\parallel$) to the principal molecular axis as well as isotropic $g$-values for the minimum structures $D_{3d}$, $C_{2h}$ and $D_{4h}$ are compared to the experimental values (exp.)
[b]Transition state structure connecting two equivalent $D_{3d}$ symmetric structures
[c]Not a stationary point
[d]$g_{iso} = (2{\cdot}g_\perp + g_\parallel)/3$

assumption, we conducted EPR measurements of **1** at 4 K. At this temperature, the spectrum indicates axial anisotropy with two principle $g$-values of $g_\parallel = 1.953$ and $g_\perp = 2.156$, being consistent with an axial $D_{3d}$-symmetry of the $[Cr(CO)_6]^{\bullet+}$ cation (Fig. 2d).

**Ab initio and DFT investigations.** To complete the answer to the question of the correct point group of the $[Cr(CO)_6]^{\bullet+}$ cation, we performed symmetry restricted structure optimizations at the TPSSh/def2-TZVPP level of theory. In a comparative study, we found the TPSSh functional to yield reasonable structural parameters (Supplementary Table 4). The so-obtained structures were used for DLPNO-CCSD(T) single-point electronic energy calculations. In addition, we calculated $g$-tensors for minimum structures by means of the NEVPT2-SA-CAS-SCF method. The results are summarized in Table 4.

In accordance with an extensive study[51] of the isoelectronic V$(CO)_6$ complex, we found the $D_{3d}$ structure to be the lowest in energy. The $D_{2h}$ structure is a transition state connecting two equivalent $D_{3d}$ minima. Alike, the $O_h$ symmetric structure is not a stationary point and convergence in the SCF procedure can only be achieved by allowing (unphysical) broken occupation numbers in the unrestricted treatment. Similar to V$(CO)_6$[57], we identified a $C_{2h}$ symmetric structure, which shows only a very small structural and energetic deviation from the $D_{3d}$ one. We note that the

calculated minimal relative energy of the $D_{2h}$ transition state ($+0.85\,kJ\,mol^{-1}$) is in perfect agreement with the EPR observation of a fluxional structure at $100\,K$. Thermal energy corresponds to $0.83\,kJ\,mol^{-1}$ at $100\,K$. Only at considerably lower temperature ($4\,K$) the structure is static. The $D_{4h}$ symmetric structure is a local minimum notably higher in energy and seems only to be accessible on the potential energy surface by passing the $D_{2h}$ transition state to higher energies. Furthermore, the CAS-SCF calculations yield a much more anisotropic g-tensor for the $D_{4h}$ structure than for the $D_{3d}$ one. Hence, even though there is some deviation, the comparison of calculated and experimental g-values clearly confirms the $[Cr(CO)_6]^{\bullet+}$ cation to have a $D_{3d}$ symmetric ground state structure. Thus, the symmetry taken by the $[Cr(CO)_6]^{\bullet+}$ cation in the crystal structure—best realized in the cubic structure **2** with an isolated Cr–CO moiety residing on a $\bar{3}$-position of local $D_{3d}$ site symmetry—is really rooted in its electronic structure and not caused by interactions with the anion or crystal packing effects. This is in agreement with our claim that the WCA $[F-\{Al(OR^F)_3\}_2]^-$ is one of the least coordinating anions known[58] and as such truly introduces pseudo-gas phase conditions in the solid state[59].

## Discussion

The right combination of $[NO]^+$ as oxidant with the WCAs $[Al(OR^F)_4]^-$ and $[F-\{Al(OR^F)_3\}_2]^-$ and the unique reactivity of $Cr(CO)_6$ gives facile access to the unexplored field of homoleptic 17 VE radical carbonyl cations, as well as heteroleptic 18 VE chromium carbonyl/nitrosyl cations. $[Cr(CO)_6]^{\bullet+}[WCA]^-$ shows surprising stability and is closely related to the isoelectronic and isostructural $V(CO)_6$ with the same $D_{3d}$ symmetric ground state, which is supported by experimental IR, Raman, and EPR spectroscopic investigations, as well as magnetic measurements and calculated DFT/full ab initio data. Apart from its textbook importance, $[Cr(CO)_6]^{\bullet+}$ might be a valuable 17 VE metalloradical precursor for substitution chemistry, which is not part of this report. Especially, follow-up chemistry and take up by others is facilitated by the access to **1** and **2** using standard glassware, organic solvents, and commercially available starting materials. We expect transfer of this methodology to other transition metal carbonyls. Of those, the group 6 metal carbonyls molybdenum and tungsten are currently in our focus.

## Methods

**General**. Full details of the employed methods and additional information are given in Supplementary Information: pictures of the glassware used and the compounds themselves (Supplementary Figures 1–4); synthesis and spectra of NO[F-{Al(OR^F)_3}_2] (Supplementary Figures 5–8); EPR spectroscopy (Supplementary Figure 37, Supplementary Tables 4 and 5, Supplementary Data 1); calculated geometries (Supplementary Figures 76 and 77, Supplementary Data 8–11), simulated spectra (Supplementary Figures 38–41) and calculated energies (Supplementary Figure 75, Supplementary Tables 10 and 11); UV/Vis spectra (Supplementary Figure 50); crystal structures (Supplementary Figures 67–72, Supplementary Data 2–7, Supplementary Tables 8 and 9) and Hirshfeld plots (Supplementary Figures 73 and 74). For space restrictions, here we only report the data of the [Al(OR^F)_4]^- salts. However, all experimental details for the [F-{Al(OR^F)_3}_2] salts and their spectra are given as well (Supplementary Figures 13–16, 22–26 (NMR), Supplementary Figures 44, 45, 48, 49 (IR/Raman)). For all compounds, pXRD measurements including Rietveld refinements strongly support their phase pure nature (Supplementary Figures 51–66). Elemental analyses proved to be unreliable for salts with these kinds of anions; a more thorough discussion is given in the Supplementary Information (S.I. section 12 and Supplementary Table 6).

**Synthesis of $[Cr(CO)_6][Al(OR^F)_4]$ (1)**. A double-Schlenk flask was equipped inside the glove box with $Cr(CO)_6$ (51 mg, 0.232 mmol, 1 eq.) and $NO[Al(OR^F)_4]$ (230 mg, 0.231 mmol, 1 eq.). Then, $CH_2Cl_2$ (~4 mL) was condensed at $-196\,°C$ onto the reaction mixture and it was allowed to thaw to $-78\,°C$ under dynamic vacuum. Upon dissolution, the color turned initially blue and a gas evolution was visible. The mixture was stirred under dynamic vacuum (so that the solvent was very mildly boiling) until it reached r.t. over the period of an hour. The $CH_2Cl_2$ (~2 mL) was then filtered off and the remaining solid was dried in vacuo in order to yield **1** as an off-white to beige solid (260 mg, 0.219 mmol, 94%). Single crystals were obtained by slow vapor diffusion of n-pentane in a solution of **1** in 1,2-$F_2C_6H_4$ (oDFB) at r.t. FTIR (ZnSe, ATR): $\tilde{\nu}/cm^{-1}$ (intensity) = 2094 (s), 1843 (vvw, trace of $[Cr(CO)_5(NO)]^+$), 1508 (vvw, trace of residual oDFB), 1352 (vw), 1298 (mw), 1272 (ms), 1239 (s), 1208 (vvs), 1161 (ms), 969 (vvs), 831 (vw), 756 (vvw), 726 (vvs), 571 (vvw), 560 (vw) (Supplementary Figure 42). FT Raman (1000 scans, 100 mW, $4\,cm^{-1}$): $\tilde{\nu}/cm^{-1}$ (intensity) = 2175 (vvs), 2128 (mw), 2062 (vw), 1304 (vvw), 1272 (vvw), 1235 (vvw), 1163 (vvw), 977 (vvw), 798 (w), 747 (w), 572 (vvw), 563 (vvw), 538 (vw), 368 (vvw), 332 (m), 288 (w), 234 (vvw), 173 (vvw) (Supplementary Figure 43). $^1H$ NMR (400.17 MHz, oDFB, 298 K): only solvent signals; $^{13}C\{^1H\}$ NMR (100.62 MHz, oDFB, 298 K): $\delta/ppm = 121.7$ (q, $^1J(C,F) = 293\,Hz$, 12C, $CF_3$, in part overlapped by solvent signals); $^{19}F$ NMR (376.54 MHz, oDFB, 298 K): $\delta/ppm = -75.3$ (s, 36F, $4 \times C(CF_3)_3$); $^{27}Al$ NMR (104.27 MHz, oDFB, 298 K): $\delta/ppm = 35.0$ (s, 1Al, $Al(OR^F)_4$) (Supplementary Figures 9–12).

**Synthesis of $[Cr(CO)_5(NO)][Al(OR^F)_4]$ (3)**. A double-Schlenk flask was equipped inside the glove box with $Cr(CO)_6$ (51 mg, 0.232 mmol, 1 eq.) and $NO[Al(OR^F)_4]$ (228 mg, 0.229 mmol, 1 eq.). Then, $CH_2Cl_2$ (~3 mL) was added at r.t. to the reaction mixture and the closed vessel was stirred for 14 days at r.t. After drying in vacuo, **3** yielded as an orange solid (250 mg, 0.210 mmol, 92%). Orange single crystals were obtained by slow vapor diffusion of n-pentane in a solution of **3** in oDFB at r.t. FTIR (ZnSe, ATR): $\tilde{\nu}/cm^{-1}$ (intensity) = 2184 (vw), 2164 (vvw), 2127 (vvw), 2108 (ms), 2074 (vvw), 1841 (mw), 1821 (vvw, trace of unknown impurity), 1353 (vw),1299 (mw), 1273 (ms), 1251 (ms),1239 (ms), 1210 (vvs), 1162 (vvw), 1140 (vw), 971 (vvs), 866 (vvw), 832 (vw), 756 (vvw), 727 (vs), 657 (w), 640 (vw), 571 (vvw), 561 (vw) (Supplementary Figure 46). FT Raman (100 scans, 500 mW, $4\,cm^{-1}$): $\tilde{\nu}/cm^{-1}$ (intensity) = 2185 (m), 2175 (vw), 2164 (m), 2127 (vvs), 2112 (vw), 2076 (vvw), 1843 (vvw), 1304 (vvw), 1275 (vvw), 977 (vvw), 797 (vw), 747 (vw), 640 (vw), 562 (vvw), 539 (vvw), 487 (vvw), 369 (vvw), 332 (vw), 290 (vvw), 234 (vvw) (Supplementary Figure 47). $^1H$ NMR (300.18 MHz, oDFB, 298 K): only solvent signals; $^{13}C\{^1H\}$ NMR (100.62 MHz, oDFB, 298 K): $\delta/ppm = 201.9$ (s, $4C_{equatorial}$, $Cr(CO)_5(NO)$), 187.3 (s, $1C_{axial}$, $Cr(CO)_5(NO)$), 121.7 (q, $^1J(C,F) = 293\,Hz$, 12C, $CF_3$, in part overlapped by solvent signals), 79.5 (m, 4C, $4 \times C(CF_3)_3$); $^{14}N$ NMR (21.69 MHz, oDFB, 298 K): $\delta/ppm = 17$ (br. s, 1N, $Cr(CO)_5(NO)$); $^{19}F$ NMR (376.54 MHz, oDFB, 298 K): $\delta/ppm = -75.3$ (s, 36F, $4 \times C(CF_3)_3$; $^{27}Al$ NMR (78.22 MHz, oDFB, 298 K): $\delta/ppm = 35.0$ (s, 1Al, $Al(OR^F)_4$) (Supplementary Figures 17–21).

## Data availability

Atomic coordinates and structure factors for the crystal structure of **1–4** are deposited at the Cambridge Crystallographic Data Centre (CCDC) under the accession codes 1844666 (compound **2**), 1844667 (**1**), 1844668 (**3**), 1844669 (**4**), 1886541 (NO disorder model for **3**), and 1886542 (NO disorder model for **4**); copies of the data can be obtained free of charge from www.ccdc.cam.ac.uk/data_request/cif. All the other data that support the findings of this study are available within Supplementary Information files, and are available from the corresponding authors on reasonable request.

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

## Acknowledgements

The authors would like to thank Dr. Thilo Ludwig for powder-XRD measurements, Dr. Harald Scherer for NMR measurements and the Bruker BioSpin GmbH for conducting

low temperature EPR measurements. J.B. gratefully acknowledges financial support by the LGFG Graduate Funding. W.F. gratefully acknowledges the Carl-Zeiss-Stiftung for financial and the Studienstiftung des Deutschen Volkes e.V. for general support. This article is dedicated to Prof. Dr. Hansjörg Grützmacher on occasion of his 60th birthday.

## Author contributions

I.K. supervised the project. J.B. designed and performed the experiments and all characterization but EPR measurements. D.H. carried out the general quantum mechanical calculations. M.D. did the Rietveld-refinement of the pXRD data, F.B. supervised the EPR measurements and respective EPR calculations carried out by W.F. All authors contributed to the discussion of the results and preparation of the manuscript.

## Additional information

**Competing interests:** The authors declare no competing interests.

