## [Peer Review File · Nature Communications]

Reviewers' comments:

Reviewer #1 (Remarks to the Author):

In caption Figure 2, this sentence makes no sense.

Inclusion of a disordered nitrogen atom into the refinement or not did only slightly change the agreement factors.

Structures:

I realize that they modelled the NO complex identical to the CO₆ complex, but they need to adjust the formula and make sure the density is correct.

I would have like to see them model the nitrogen partial occupancy on the NO complexes. This would make for a better overall real result.

These structures are of course difficult to distinguish crystallographically, but I would question the validity of the results based on the esd's alone. Would be nice to see a picture of the crystals used, possible on modern diffractometers. Also, the smallest crystal provided the most intense data. (P4n system)

It would be nice to know if other crystals were tested and a smaller cell versus lager cell seen in all the crystals. The Hirschfeld results suggest that one of the CO could be replace (two over symmetry) preferentially based on charges, does this correspond to the shortest Cr-c bond? Again, attempts at what was tried in replacing the carbon and nitrogen system not well explained and therefore not sure if some of the obvious disordered models were attempted for refinement.

Do the two refinements behave the same if you interchange cell dimensions with the hkl files?

The comparison of the powder diffraction to calculated is nice, but it might be interesting to see if there are differences between the all CO and NO complex.

Reviewer #2 (Remarks to the Author):

Outstanding research which definitely establishes for the first time isolable examples of a paramagnetic homoleptic metal carbonyl cation, [\bullet Cr(CO)₆](+) and a mixed cationic compound containing only CO and NO as ligands [Cr(CO)₅NO](+). These are landmark developments in metal carbonyl/nitrosyl chemistry! I will be extraordinarily happy to proclaim these as "molecules of the week" in my organometallic chemistry lectures. First class novelty!

One issue: the abstract suggests that Cr(CO)₆(+) is the first isolable chromium carbonyl cation. However, the first one was the heteroleptic [CpCr(CO)₄](+), isolated by E.O. Fischer and K. Ulm, as a tetrafluoroborate salt, and well characterized by its magnetic susceptibility, IR spectra and a full elemental analysis (oxygen by difference; Z. Naturforschg, (1961) 16b, 757. (I am sure the authors knew of this species) Simply modify the abstract to correct.

Another and more compelling issue: although the key compounds are well-characterized by a variety of physical techniques and on this basis there is absolutely no reason to question their existence, unfortunately, I see no elemental analyses on bulk samples. Thus, in the syntheses of compounds (1) and (3), isolated yields of 94% and 92%, respectively, are claimed, but there are no elemental analyses ,EA's, presented in support of these yields. Fully consistent spectral data are presented in support of these formulations, but, of course, bulk samples often contain spectroscopic "silent" impurities. If there is some reason why EA's could not be obtained or provided less than satisfactory results, this should be indicated in the Experimental. Otherwise, one could worry greatly about the reproducibility of the syntheses. Most certainly these preps could not be published in Inorganic Syntheses without these data.

Reviewer #3 (Remarks to the Author):

Comments on the paper “Stable Salts of the Hexacarbonyl Chromium(I) Cation and its Pentacarbonyl-Nitrosyl Chromium(I) Analogue” by Bohnenberger et al.

The authors report about a joint experimental and theoretical study which gives evidence of the first homoleptical carbonyl radical cation of a transition metal that has been isolated in a condensed phase. There is further work about the related nitrosyl cation complex $[\text{Cr}(\text{CO})_5(\text{NO})]^+$. The results are very interesting for the broad community and I recommend publication of this work. However, I request that the authors revise their work and write it in a more digestible way. The present work is a bit sloppily written and should be revised according to the following points.

1. Figure 1 shows some experimental and theoretical spectra which are difficult to entangle. On top of the figure are the infrared and Raman spectra and it is not obvious which of the calculated and experimental work belongs to the IR and which to the Raman spectra. Also, a figure should be readable without that the text is studied. At what levels were the calculated frequencies obtained? I noticed that for the chromium hexacarbonyl cation there are two signals from the calculations and I wonder which of the experimental signals they refer to. The same applies to the pentacarbonyl-nitrosyl cation on the right-hand side.
2. On page 2, the authors make references to “most carbonyl complexes” when they in reality mean only transition metal complexes that have been isolated in a condensed space. A lot of main-group complexes and numerous carbonyl complexes in the gas phase do not obey the 18 electron rule. The authors should be a bit more precise in their statements.
3. The authors make a statement about cations, which have been synthesized in the 1990s. The statement on page 2 is made without any reference. There have been theoretical and experimental studies of these systems, which should be mentioned.
4. The final sentence of the full paragraph on page 2 is incomprehensible: “Here also on the chromium hexacarbonyl cation $[\text{Cr}(\text{CO})_6]^{\bullet+}$, which is of interest to this report.”
5. Table 2 on page 5 gives theoretical and experimental results of the vanadium species. It is not clear if these data are coming from the literature or if the authors have studied these systems. If so, I wonder why the vanadium species were calculated with BP86 whereas the present chromium systems have apparently been calculated at the TPSSH level.
6. Table 4 gives some relative electronic energies without saying from where they were taken. They are probably calculated values. If so, at which level of theory have they been obtained?

In summary, the work reports about chemically interesting systems, but the paper sloppily written. The authors should do a more sincere work in presenting their results.

Response to the Referees

The main concern of the referees was to demonstrate the clear identity and phase-purity of the samples, especially since the pathway to the salts is the same using NO^+ as reagent and only the conditions (low or ambient temperature) decide on the outcome. We have done our best to make these points clear and we would like to take the opportunity to summarize the main points here that allow differentiating between the all-carbonyl and the mixed nitrosyl-carbonyl complexes.

IR/Raman: This bulk method addresses crystalline and non-crystalline domains. The presence or absence of the NO-stretch greatly changes the pattern and even very tiny impurities of the mixed nitrosyl-carbonyl complex are visible. Let me demonstrate this with two figures. The first shows the regular IR spectra of the two pure NO complexes **3** and **4**.

You notice the very intense NO-stretch at about $1842 / 1843 \text{ cm}^{-1}$. In the next set of IR spectra of the (almost completely) pure all-carbonyl compounds **1** and **2** it is easy to notice that this intense NO stretch is virtually absent. A very tiny NO stretch of a very small contamination of the compounds **1** and **2** with the respective salts **3** and **4** is evident (see the box).

Thus, already from the comparison of the vibrational spectra of the bulk materials it is evident that the materials have a purity exceeding 95 %. Note that in the supplemental information, a large Table with the full assignment of all (!)

bands in the spectra is included. Working since many years with these anions gives us the long-term experience and knowledge to assign all anion bands and therefore allow for a complete assignment of all cation bands.

In addition, the supplemental information in section 8 also lists the gas phase IR spectra of the gas phase over the reaction mixtures leading, under mild conditions, only to NO gas and, upon equilibration at room temperature, only to CO gas. This is in complete agreement with the outcome of all other analyses and demonstrates that temperature and time control allows differentiating between the kinetic product (all-carbonyl complexes) and the thermodynamic mixed NO-carbonyl complexes. For your convenience, we include the spectra here:

Figure. Gas phase IR spectra of the reaction mixtures from the synthesis of $[\text{Cr}(\text{CO})_6]^+[\text{Al}(\text{OR}^{\text{F}})_4]^-$ after 20 min (bottom spectrum, red) and the synthesis of $[\text{Cr}(\text{CO})_5(\text{NO})]^+[\text{Al}(\text{OR}^{\text{F}})_4]^-$ after 14 d (top spectrum, blue). Taken from Section 8 of the Supplemental Information.

Colours: The colours of the all-carbonyl and the mixed NO-carbonyl compounds are very different. Note that also UV-Vis spectra of all batches are included in the supplemental information. This easily allows differentiation between the two sets of compounds (see also picture in the point-to-point answer).

Magnetic and EPR measurements: Only the all-carbonyl compounds are open shell. The VT-EPR spectra provide proof for the presence of the assignment as chromium hexacarbonyl radical cation. More important with respect to the purity of the materials are, however, the bulk magnetic measurements of both compounds **1** and **2**. For both compounds, the measurements reveal the presence of an effective magnetic moment μ_{eff} of 2.04 / 2.06 μ_{B} . Any impurity of the diamagnetic mixed NO-complex would lower these values considerably. This speaks against noticeable cross-contamination. See Section 6 of the supplemental Information.

scXRD and pXRD measurements: At first let me reiterate that we only used material for these measurements that by IR and Raman measurements was shown to be (almost completely) pure (see spectra above). In addition, all crystals had the respective colours (pale yellow for the all carbonyl-version and red-orange for the mixed NO-carbonyl case). Since Cr-N distances in the mixed complexes are expected to be shorter (see QM), also their unit cell is slightly, but noticeably smaller than that of the all-carbonyl compounds. To show this also for the microcrystalline bulk, we performed Rietveld-refinements of the powder data recorded at the same temperature like the single crystal data (100 K). Since the pXRD recorded higher angle data than the scXRD data, the resolution is better and the standard deviations are further reduced. Thus, also the bulk of the material (and not just an isolated single crystal!) show the smaller lattice parameters for the mixed NO-carbonyl complexes.

This shall be exemplarily shown here for the volumes of $[\text{Al}(\text{OR}^{\text{F}})_4]^-$ salts **1** and **3**: The cell volumes are $V = 1776.42(4)$ (**1**, all carbonyl) and $V = 1767.39(3)$ (**3**, mixed-NO-carbonyl). With a standard deviation of only 0.04 and 0.03 \AA^3 (the second digit after (!) the comma), the volume difference of 9.03 \AA^3 between both salts is more than statistically relevant and shows that these are independent and different materials. More details in the point-to-point answer as well as the supplemental information, section 12.

Elemental analyses: Our personal experience speaks against them. Due to the high fluorine content (about 57% and 61%), reliable elemental analyses are problematic (see for example: Marcó, A.; Compañó, R.; Rubio, R.; Casals, I.

Microchim. Acta 2003, 142, 13–19 or the *Organometallics* editorial *Organometallics* 2016, 35 (19), 3255–3256) and we refrained from using them. In addition, we have done quite a few tests with our materials and using the equipment available to us in the chemistry department in Freiburg. Unfortunately, also with the new set up that was acquired only 3 years ago and supposedly able to treat CF₃ groups, the test elemental analyses of electrochemically and spectroscopically extremely pure air- and water-stable NBu₄⁺[Al(OR^F)₄]⁻ gave combustion analysis values that were off by erratically 2-3 % for samples from the same batch. However, the facility at KIT Karlsruhe is able to deal with compounds with high fluorine contents. Exemplarily, we tested a sample of [Cr(CO)₆](Al(OR^F)₄) (calcd.: C 22.26; Al 2.27; Cr 4.38; F 57.61; O 13.48) there. The results are shown in the table below.

Sample	N	C	H	S
1.178 mg	0.17	23.30	0.247	0.109
2.367 mg	0.08	22.71	0.130	0.049
Avg.	0.13	23.00	0.189	0.079
Expected [Cr(CO) ₆](Al(OR ^F) ₄)	0.00	22.26	0.000	0.000
Expected [Cr(CO) ₅ (NO)](Al(OR ^F) ₄)	1.18	21.21	0.000	0.000

The discrepancy between theoretical and experimental carbon content is small enough to be acceptable for an inorganic organometallic compound like this and is considered publishable, especially with the reasons given above. However, elemental analyses does not answer the question of the purity of the bulk materials any better than the sum of vibrational and NMR spectroscopy as well as pXRD which we thoroughly deployed in the S.I.. This is the reason, why we trust the combination of these methods more than a doubtful combustion analysis with large deviations and tolerance thereof.

Answers to Reviewer #1

comment by reviewer #1

In caption Figure 2, this sentence makes no sense.
“Inclusion of a disordered nitrogen atom into the refinement or not did only slightly change the agreement factors.”

I realize that they modelled the NO complex identical to the CO₆ complex, but they need to adjust the formula and make sure the density is correct. I would have like to see them model the nitrogen partial occupancy on the NO complexes. This would make for a better overall real result.

responses and changes made by the authors

We changed that sentence to:
“Inclusion of a disordered nitrogen atom into the refinement did only slightly change the agreement factors (Section 13, S.I.).”
 (see also the next paragraph)

This is a valid point. We are aware that a disordered NO ligand describes the ‘chemical reality’ more accurately than an ‘only CO’ refinement for the [Cr(CO)₅(NO)]⁺ system. However, based on the crystal data alone, to us it seemed not ‘scientifically correct’ to add and refine a ligand that is not differentiable by XRD means. However, we now added a refinement of a disordered NO ligand for both systems to the S.I..
 P4/n case [Cr(CO)₅(NO)](Al(OR^F)₄):
 A roughly equal distribution in the NO disorder (16%/4x17%/16%) led to the best model. The R₁ value changed from 2.52% to 2.46% when the disorder was included. A refinement of the N positions with a free variable, however, did not lead to a stable model.

The resulting bond lengths are:

w/o disorder [pm]		With NO disorder [pm]			
d(Cr-C1)	195.8(3)	d(Cr-C1)	199.7(3)	d(Cr-N1)	171(4)
d(Cr-C2)	196.2(1)	d(Cr-C2)	200.78(16)	d(Cr-N2)	170.3(16)
d(Cr-C3)	192.4(4)	d(Cr-C3)	196.4(5)	d(Cr-N3)	170(4)

Pa-3 case $[\text{Cr}(\text{CO})_5(\text{NO})][\text{F}\{\text{Al}(\text{OR}^{\text{F}})_3\}_2]$:

The inclusion of 1/6 NO led to a change of the R_1 value from 6.03% to 5.99%. However, the NO ligands resulted in a slightly tilted octahedron.

w/o disorder [pm]		With NO disorder [pm]			
d(Cr-C1)	194.9(4)	d(Cr-C1)	197.2(14)	d(Cr-N1)	187(6)

The low bond precisions for the Cr-N bonds still leaves the question, if the refinement of a disordered NO actually yields a scientifically more accurate structure model.

We decided to leave the NO data out and just report the average bond lengths in the manuscript.

(It) would be nice to see a picture of the crystals used, possible on modern diffractometers.

Unfortunately, we did not take pictures of the crystals that were used on the diffractometer. The crystals were of the same block-shape and color as the ones shown here (left: $[\text{Cr}(\text{CO})_6][\text{Al}(\text{OR}^{\text{F}})_4]$; right: $[\text{Cr}(\text{CO})_5(\text{NO})][\text{Al}(\text{OR}^{\text{F}})_4]$):

We also deposited that picture in the S.I. (chapter 2) and added it (in part) to the reaction scheme in the manuscript:

Also, the smallest crystal provided the most intense data. (P4n system)

It would be nice to know if other crystals were tested and a smaller cell versus larger cell seen in all the crystals.

The Hirschfeld results suggest that one of the CO could be replaced (two over symmetry) preferentially based on charges, does this correspond to the shortest Cr-C bond? Again, attempts at what was tried in replacing the carbon and nitrogen system not well explained and therefore not sure if some of the obvious disordered models were attempted for refinement.

Indeed, the smaller crystal of **1** with the size of 0.1x0.1x0.1 had an exposure time of 20 seconds/frame, the crystal of **3** with 0.15x0.1x0.1 dimensions was exposed 10 seconds/frame (small angles) and 15 seconds/frame (wide angles).

We did not test other crystals since we only collected the XRD data of the four different products that were purest by spectroscopy. However, see below for the powder data.

The Hirshfeld-plot in the S.I. showed only the homoleptic carbonyl complexes $[\text{Cr(CO)}_6][\text{Al(OR}^{\text{F}})_4]$ and $[\text{Cr(CO)}_6][\text{F-}\{\text{Al(OR}^{\text{F}})_3\}_2]$. We additionally calculated the plot for $[\text{Cr(CO)}_5(\text{NO})][\text{Al(OR}^{\text{F}})_4]$ below. The charged (red) surface does correspond to the Cr-C1 (195.8 pm) bond (other bond lengths: 196.2 (Cr-C2; 4-fold-Symmetry axis), Cr-C3: 192.4 pm), indicating that this is not a favored position for the disordered NO-ligand. We also added that information to the S.I., chapter 14.

Do the two refinements behave the same if you interchange cell dimensions with the hkl files?

If by that the reviewer means refining the respective $[\text{Cr}(\text{CO})_6]^+$ structure with the (smaller) cell dimensions of $[\text{Cr}(\text{CO})_5(\text{NO})]^+$: in both cases the R-values did not change. The bond lengths got very slightly smaller (although insignificant to the error margin):

d/pm	$[\text{Cr}(\text{CO})_6]$ $[\text{Al}(\text{OR}^{\text{F}})_4]$ initially	$[\text{Cr}(\text{CO})_6]$ $[\text{Al}(\text{OR}^{\text{F}})_4]$ smaller cell	$[\text{Cr}(\text{CO})_5(\text{NO})]$ $[\text{Al}(\text{OR}^{\text{F}})_4]$ (w/o disorder)
Cr-C1	196.2(3)	196.0(3)	195.8(3)
Cr-C2	199.1(1)	198.8(1)	196.2(1)
Cr-C3	196.9(3)	196.6(3)	192.4(3)
	$[\text{Cr}(\text{CO})_6]$ $[\text{F}-\{\text{Al}(\text{OR}^{\text{F}})_3\}_2]$ initially	$[\text{Cr}(\text{CO})_6]$ $[\text{F}-\{\text{Al}(\text{OR}^{\text{F}})_3\}_2]$ smaller cell	$[\text{Cr}(\text{CO})_5(\text{NO})]$ $[\text{F}-\{\text{Al}(\text{OR}^{\text{F}})_3\}_2]$ (w/o disorder)
Cr-C1	198.2(2)	192.1(2)	194.9(4)

The comparison of the powder diffraction to calculated is nice, but it might be interesting to see if there are differences between the all CO and NO complex.

A very good point. We added the respective comparisons to the S.I. with a magnification of the 3-10° and 19-25° 2Theta range to emphasize the slight difference between the two diffractograms.

Complexes **1** and **3**:

Complexes 2 and 4:

The all-CO complex (in red) indicates a slightly larger cell (smaller angles for the same reflexes). Furthermore, we did a Rietveld-refinement for a more detailed quantitative description:

Compound	[Cr(CO) ₆][Al(OR ^F) ₄]	[Cr(CO) ₅ (NO)][Al(OR ^F) ₄]	[Cr(CO) ₆][F-Al(OR ^F) ₃] ₂	[Cr(CO) ₅ (NO)][F-Al(OR ^F) ₃] ₂
Temp.	100 K	100 K	100 K	100 K
Space Group	P4/n	P4/n	Pa $\bar{3}$	Pa $\bar{3}$
Cell	a =	a =	a =	a =
	13.65605(15)	13.63318(11)	17.27593(8)	17.26357(8)
	c =	c =		
	9.52568(17)	9.50905(12)		
V =	V =	V =	V =	V =
	1776.42(4)	1767.39(3)	5156.13(7)	5145.08(7)
R _f ²	0.0403	0.0428	0.0456	0.0457
wR _p	0.0363	0.0408	0.0309	0.0341
R _p	0.0285	0.0314	0.0243	0.0266
χ^2	2.690	4.038	2.564	2.890

The Rietveld refinement clearly gave the statistically significant larger cells for the all-carbonyl compounds and the smaller cells for the mixed-NO-CO compounds. In addition to the color differentiation, the clear IR and Raman evidence for the nature of both compounds, this is further clear support for the correct assignment of the structures and proves the composition of the bulk material.

Answers to Reviewer #2

comment by reviewer #2

One issue: the abstract suggests that $\text{Cr}(\text{CO})_6(+)$ is the first isolable chromium carbonyl cation.

However, the first one was the heteroleptic $[\text{CpCr}(\text{CO})_4](+)$, isolated by E.O. Fischer and K. Ulm, as a tetrafluoroborate salt, and well characterized by its magnetic susceptibility, IR spectra and a full elemental analysis (oxygen by difference; Z. Naturforsch, (1961) 16b, 757. (I am sure the authors knew of this species) Simply modify the abstract to correct.

Another and more compelling issue: although the key compounds are well-characterized by a variety of physical techniques and on this basis there is absolutely no reason to question their existence, unfortunately, I see no elemental analyses on bulk samples. Thus, in the syntheses of compounds (1) and (3), isolated yields of 94% and 92%, respectively, are claimed, but there are no elemental analyses, EA's, presented in support of these yields. Fully consistent spectral data are presented in support of these formulations, but, of course, bulk samples often contain spectroscopic "silent" impurities. If there is some reason why EA's could not be obtained or provided less than satisfactory results, this should be indicated in the Experimental. Otherwise, one could worry greatly about the reproducibility of the syntheses.

responses and changes made by the authors

We modified the abstract to clarify this:

*"They are the first stable salts of a homoleptic carbonyl radical cation, as well as the first **homoleptic** chromium carbonyl cations in condensed phases."*

First, we deployed powder-XRD plots without background correction in the S.I. (chapter 12) to emphasize the absence of (large amounts) of amorphous impurities:

Complex 1:

Complex 2:

Complex 3:

Complex 4:

Furthermore, we provided a second picture of the obtained crystals supporting good yields of crystalline product (see reviewer#1, statement 3). The crystallization of an exemplary reaction yielded at least 88% of large block-shaped crystals as seen in the picture mentioned.

Regarding elemental analyses: Please see the comment in the introductory statement of this answer letter.

comment by reviewer #3

Figure 1 shows some experimental and theoretical spectra which are difficult to entangle. On top of the figure are the infrared and Raman spectra and it is not obvious which of the calculated and experimental work belongs to the IR and which to the Raman spectra. Also, a figure should be readable without that the text is studied. At what levels were the calculated frequencies obtained? I noticed that for the chromium hexacarbonyl cation there are two signals from the calculations and I wonder which of the experimental signals they refer to. The same applies to the pentacarbonyl-nitrosyl cation on the right-hand side.

responses and changes made by the authors

We added additional information to Figure 1 to clarify which vibrational spectrum is calculated and experimental (apart from coloring them in the same fashion).

The full text that refers to Figure 1 explains the discrepancy between the ideal D_{3d} symmetry of the calculated $[\text{Cr}(\text{CO})_6]^+$ cation and the experimental spectra. The energetic ground state is a D_{3d} symmetric one, which can fluctuate at room temperatures due to a low-lying energy barrier. This results in the experimental spectrum with one sharp (the all-symmetric CO-stretch is unaffected by this slight changes in geometry) and one very broad Raman band. Naturally, this fluctuation cannot be described by a simulated spectrum of the D_{3d} ground state. We clarified this by rewriting the statement in the full text to:

"The respective Raman spectra show a sharp band at about 2174 cm^{-1} for the all-symmetric stretch vibration as well as a broad band centred around 2127 cm^{-1} . The position and shape of these bands are in agreement with the Raman spectrum of the isoelectronic but neutral D_{3d} - $\text{V}(\text{CO})_6$ (cf. Table 1, and S.I., Figure 5)⁵¹. A Jahn-Teller induced fluxionality at room temperature leads to the broad band at about 2127 cm^{-1} not affecting the all-symmetric stretch mode. This fluxionality freezes out in the vanadium case only at temperatures below 16 K. At higher temperatures, a very low-lying transition state probably allows for equilibration – even on the fast time scale of vibrational spectroscopy, leading to the discrepancy between experimental and the D_{3d} -simulated spectrum."

We also rewrote the caption of Figure 1 for clarity to:

"Block figure showing essential IR, Raman and EPR spectra of complexes 1–4. Top: Stacked IR and Raman spectra of compounds 1 (dark blue), 2 (red), 3 (blue), 4 (purple) and the respective simulated calculated spectra (black, BP86-D3BJ/def2-TZVPP) of the cations in the CO / NO stretching range between 1800 and 2300 cm^{-1} "

And:

"f) Equilibration path that transforms at 100 K the two D_{3d} ground states over a low-lying D_{2h} transition state, yielding a coalescent signal in the EPR spectrum."

We understand that since we decided not to use scaling factors for the calculated spectra, it can be difficult to see immediately, which band belongs to which. Especially, since phenomena such as removal of the degeneracy of modes (IR: E mode at 2107 cm^{-1}) due to slightest cation/anion interactions and ^{13}C -isotope shifts (IR and Raman: E mode at 2074 cm^{-1}) for $[\text{Cr}(\text{CO})_5(\text{NO})]^+$ cannot be described by one simulated spectrum.

In order to compensate for that, we gave the full assignment of all the vibrational spectra in Table 2.

On page 2, the authors make references to “most carbonyl complexes” when they in reality mean only transition metal complexes that have been isolated in a condensed space. A lot of main-group complexes and numerous carbonyl complexes in the gas phase do not obey the 18 electron rule. The authors should be a bit more precise in their statements.

The authors make a statement about cations, which have been synthesized in the 1990s. The statement on page 2 is made without any reference. There have been theoretical and experimental studies of these systems, which should be mentioned.

The final sentence of the full paragraph on page 2 is incomprehensible: “Here also on the chromium hexacarbonyl cation $[\text{Cr}(\text{CO})_6]^{+\bullet}$, which is of interest to this report.”

Table 2 on page 5 gives theoretical and experimental results of the vanadium species. It is not clear if these data are coming from the literature or if the authors have studied these systems. If so, I wonder why the vanadium species were calculated with BP86 whereas the present chromium systems have apparently been calculated at the TPSSH level.

Table 4 gives some relative electronic energies without saying from where they were taken. They are probably calculated values. If so, at which level of theory have they been obtained?

We changed the respective part in order to clarify that we focus our statements to homoleptic transition metal complexes in the condensed phase only:

“Most condensed phase homoleptic transition metal carbonyl complexes and all neutral mononuclear homoleptic transition metal carbonyl complexes in particular, obey the 18-electron rule⁷; the only exception is $\text{V}(\text{CO})_6$ as a 17 valence electron (VE) species.”

The literature references are found in Table 1 and since the initial submission guide states that references should typically be limited to 50, it is difficult to account for all original citations. Therefore, we have taken the decision, to refer to review articles. Thus in the footnote to Table 1 is explicitly stated: **TMCCs are only referenced, if not mentioned in the reviews.^{19–22}** However, we now additionally properly referenced the syntheses and characterization of the respective cations in the full text for clarity. For space reasons again the focus is laid upon synthetic work only.

“Transition metal carbonyl cations (TMCCs), however, could not be accessed until about 1960, when the octahedral carbonyl cation $[\text{Mn}(\text{CO})_6]^+$ was discovered¹⁰. Since the 1990s, mainly superacidic media enabled the synthesis, isolation and full characterization of several homoleptic TMCCs such as $[\text{Au}(\text{CO})_2]^+$ ^{11–13}, $[\text{Fe}(\text{CO})_6]^{2+}$ ¹⁴, $[\text{Co}(\text{CO})_5]^+$ ¹⁵ or even superelectrophilic $[\text{Pd}(\text{CO})_4]^{2+}$ ¹⁶ or $[\text{Ir}(\text{CO})_6]^{3+}$ ¹⁷.”

We replaced the last two sentences of this paragraph:

“However, the chromium hexacarbonyl cation, as a prototype example for such open-shell TMCC systems, was the subject of several electrochemical investigations^{30–33}, as well as gas phase and theoretical studies^{34–36}.”

Note, that in this case we also had to restrain the citations to relevant gas phase, electrochemical and theoretical publications.

Table 2 gives only experimental data from previous works on $\text{V}(\text{CO})_6$ and $\text{V}(\text{NO})(\text{CO})_5$, which we properly referenced now and added an “Exp.” for clarity. The calculations shown in the Table 2 to refer to the complexes of this work, which was visually clarified by adding a separation double-line in Table 2.

The calculations on vibrational spectra were done with BP86 as indicated in the caption of Table 2. TPSSH proved to be more accurate for bond lengths and the EPR simulations but not for vibrational spectroscopy and was therefore used for the prior two only.

We added the missing information on the calculations to Table 4:

“Calculated (DLPNO-CCSD(T)/def2-TZVPP) relative electronic energies E in cm^{-1} (kJ mol^{-1}) of structures (D3BJ-TPSSH/def2-TZVPP) with different point groups for the $[\text{Cr}(\text{CO})_6]^{+\bullet}$ cation. The calculated (NEVPT2-SA-CAS-SCF/cc-pVTZ) anisotropic g -tensor components perpendicular g_{\perp} and parallel g_{\parallel} to the principal molecular axis as well as

isotropic g-values for the minimum structures D_{3d} , C_{2h} and D_{4h} are compared to the experimental values (Exp.) “

REVIEWERS' COMMENTS:

Reviewer #1 (Remarks to the Author):

The authors have gone to great length to convince us that they have made the now not as novel, complex. I have no doubt that they have indeed made the material in question, but I am still bothered that they want to report the crystal structure of Cr(CO)₆ and not the actual (Scientifically correct) Cr(CO)₅NO complex that they are modeling. It is obvious that the model will not provide bond lengths and angles that are absolute for the disorder, but this occurs in all disordered models.

As a crystallographer can not accept this structure in the paper reporting the wrong atoms for the formula. This data will be deposited in the data base as a Cr(CO)₆ complex when it is a Cr(CO)₅(NO)complex.

Reviewer #2 (Remarks to the Author):

Comments concerning difficulty in obtaining satisfactory EA values and best attempts to do so should be included. In all other respects the manuscript will be a fine contribution to Nature Comm.

Reviewer #3 (Remarks to the Author):

The authors have responded satisfactorily to my commentaries. The revised manuscript can be accepted for publication.

Reviewer #1

comment by reviewer #1

The authors have gone to great length to convince us that they have made the now not as novel, complex. I have no doubt that they have indeed made the material in question, but I am still bothered that they want to report the crystal structure of Cr(CO)₆ and not the actual (Scientifically correct) Cr(CO)₅NO complex that they are modeling. It is obvious that the model will not provide bond lengths and angles that are absolute for the disorder, but this occurs in all disordered models.

As a crystallographer can not accept this structure in the paper reporting the wrong atoms for the formula. This data will be deposited in the data base as a Cr(CO)₆ complex when it is a Cr(CO)₅(NO)complex.

responses and changes made by the authors

We now also deposited the structure models with NO disorder in the CCDC.

The accession numbers were added to the manuscript as well as the full crystallographic information to the Supplementary Information.

Reviewer #2

comment by reviewer #2	responses and changes made by the authors
Comments concerning difficulty in obtaining satisfactory EA values and best attempts to do so should be included. In all other respects the manuscript will be a fine contribution to Nature Comm.	We now added a short note to Elemental Analyses in the Experimental Section of the manuscript, referring to the Supplementary Information (Section 13) where the full discussion (that was also showcased in the last “Authors Response”) is deployed.

Reviewer #3

comment by reviewer #3

responses and changes made by the authors

The authors have responded satisfactorily to my commentaries. The revised manuscript can be accepted for publication.